# Exploring options for reprocessing of N95 Filtering Facepiece Respirators (N95-FFRs) amidst COVID-19 pandemic: A systematic review

Diptanu Paul[☉], Ayush Gupta[iD]*[☉], Anand Kumar Maurya[‡]

Department of Microbiology, All India Institute of Medical Sciences, Bhopal, Madhya Pradesh, India

☉ These authors contributed equally to this work.
‡ These author also contributed equally to this work.
* ayush.microbiology@aiimsbhopal.edu.in

**Data Availability Statement:** All relevant data are within the manuscript and its Supporting Information files.

**Funding:** The authors received no specific funding for this work.

## Abstract

### Background

There is global shortage of Personal Protective Equipment due to COVID-19 pandemic. N95 Filtering Facepiece Respirators (N95-FFRs) provide respiratory protection against respiratory pathogens including SARS-CoV-2. There is scant literature on reprocessing methods which can enable reuse of N95-FFRs.

### Aim

We conducted this study to evaluate research done, prior to COVID-19 pandemic, on various decontamination methods for reprocessing of N95-FFRs.

### Methods

We searched 5 electronic databases (Pubmed, Google Scholar, Crossref, Ovid, ScienceDirect) and 1 Grey literature database (OpenGrey). We included original studies, published prior to year 2020, which had evaluated any decontamination method on FFRs. Studies had evaluated a reprocessing method against parameters namely physical changes, user acceptability, respirator fit, filter efficiency, microbicidal efficacy and presence of chemical residues post-reprocessing.

### Findings and conclusions

Overall, we found 7887 records amongst which 17 original research articles were finally included for qualitative analysis. Overall, 21 different types of decontamination or reprocessing methods for N95-FFRs were evaluated. Most commonly evaluated method for reprocessing of FFRs was Ultraviolet (Type-C) irradiation (UVGI) which was evaluated in 13/17 (76%) studies. We found published literature was scant on this topic despite warning signs of pandemic of a respiratory illness over the years. Promising technologies requiring expeditious evaluation are UVGI, Microwave generated steam (MGS) and based on Hydrogen

**Competing interests:** The authors have declared that no competing interests exist.

peroxide vapor. Global presence of technologies, which have been given Emergency use authorisation for N95-FFR reprocessing, is extremely limited. Reprocessing of N95-FFRs by MGS should be considered for emergency implementation in resource limited settings to tackle shortage of N95-FFRs.

### Systematic review identifier

PROSPERO, PROSPERO ID: CRD42020189684, (https://www.crd.york.ac.uk/prospero/display_record.php?ID=CRD42020189684).

## Introduction

Global pandemic of Corona Virus Disease of 2019 (COVID-19) has led to over 37 million cases and 1 million deaths worldwide and still counting [1]. It is caused by a novel Corona virus (nCoV), a member of family *Coronaviridae*, now renamed as SARS-CoV-2 [2]. Transmission of this virus occurs through direct, contact and airborne routes, latter particularly when aerosol generating procedures (AGPs) are done during patient care [3]. Consequently, healthcare workers (HCWs) require a full set of personal protective equipment (PPE) including gowns, gloves, facemasks, face-shields or goggles and respirators for their protection during patient care, particularly in intensive care unit settings where AGPs are done regularly [4]. This has created an unprecedented demand for PPEs leading to their global shortage forcing administrative authorities to relook the recommendations of PPE usage in a whole new light [5]. Previously, focus of PPE use strategy was not to share them between patients [6] however, due to this unprecedented crisis, it has radically shifted to optimizing the use of PPEs, their extended use and limited reuse [4, 5]. Respiratory protection is one of the fundamental rights of any employee in workplace. In healthcare settings, HCWs need to be protected against bioaerosols at all costs, which at minimum, is offered by use of N95 Filtering Facepiece Respirator (N95-FFR). These FFRs have a class of filters which is not resistant to degradation by oil and is able to remove 95% particles of 0.3 μm in size, at minimum [7]. They are single use devices ought to be discarded after use to avoid self-inoculation & cross-contamination [8].

Shortage of FFRs is not new, pangs of which were first felt during Severe Acute Respiratory Syndrome (SARS) outbreak in 2003 [9]. The possibility was also predicted for an impending Influenza pandemic consequent to which U.S. Strategic National Stockpile had plans for providing 100 million N95-FFRs nationally, but it was deemed insufficient in event of a longer pandemic [9–11]. Hence, in 2006, Institute of Medicine (IOM) constituted a committee to address reusability of facemasks. Reuse of an FFR was defined as repeatedly donning and doffing of respirator by the same wearer, with or without undergoing reprocessing in between, till it is discarded. The committee recommended reuse of respirators in the event of acute shortage provided they are not obviously damaged or soiled [11]. However, committee specified that no method exists currently for reprocessing of N95-FFRs and identified it as a research priority [11]. Consequently, various research groups began their quest to search a reprocessing method which is efficacious against respiratory pathogens, is safe for human use and maintains the integrity of various components of the respirator. Even after a decade of research, prior to COVID-19 pandemic, no method has been recommended for reprocessing of N95-FFRs. Hence, we conducted this systematic review to determine the status of research done, prior to COVID-19 pandemic, to identify technologies which can be utilized for reprocessing of N95-FFRs in present situation and can be explored in near future to tackle the global crisis of respirator shortage.

## Methods

We report this systematic review (PROSPERO ID: CRD42020189684) in accordance with the Preferred Reporting Items for Systematic Reviews and Meta-Analyses (PRISMA) guidelines [12] and checklist is provided in S1 Table.

### Search strategy

We searched five databases–Pubmed, Google Scholar, Crossref, Ovid and ScienceDirect in May 2020. Grey literature was searched using OpenGrey repository. Search strategies employing combinations of various keywords is provided in S2 Table. Searches in Google Scholar and Crossref were done using Publish or Perish 7 software (Harzing, A.W. 2007) to limit article hits and sort relevant ones. Additionally, we manually searched the back references of included studies and relevant review articles on the topic to identify further eligible studies. Articles in languages other than English were considered only when their abstracts were available in English.

### Eligibility criteria

Original research articles in any language, which evaluated a single or multiple decontamination or reprocessing methods on N95-FFRs were eligible for analysis in this study. Exclusion criteria were (i) Abstracts, posters, review articles, book chapters, letters, guidelines, point of views (ii) articles published in year 2020 and (iii) involving reprocessing or decontamination of other types of masks or respirators such as Gauze, Cloth, Spun-lace, Elastomeric and Powered-air-purifying, only.

### Data extraction

After searching all databases, we exported data in Microsoft® Excel and removed duplicates. Two reviewers (DP & AG) screened titles to remove clearly irrelevant studies. All three reviewers (AG, DP, AKM) independently screened the abstracts and full text of remaining articles to determine final eligibility and resolved any discrepancies through discussion and consensus. After included studies were finalized, data on various variables such as reprocessing method exposure variables, number, type and replicates of FFR models, parameters which were evaluated and final results was entered in Microsoft® Excel independently by all three reviewers. Extracted data was checked and analysed by one reviewer (AG) and disagreements were resolved prior to final analysis.

### Quality assessment

To assess methodological quality and risk bias of studies, a self-developed tool was designed on the basis of STROBE statement [13] due to unavailability of a validated quality assessment tool for such studies. Two authors (AKM and DP) independently assessed the methodological quality and risk bias as per tool. The scheme of scoring and grading of studies is given in S3 Table along with the final quality assessment results. Inter-author concordance on grading of studies was evaluated by third author (AG). Final quality assessment results for included studies, as shown in S3 Table, were prepared by resolving inter-author disagreements by discussion and building consensus.

# Results

## Search results

Our search strategy identified 7887 records of which 17 original research articles fit inclusion criteria for qualitative analysis [8, 14–29], methodology of the same has been described in Fig 1. No records were found in OpenGrey database using search strategy.

## Quality assessment

Of 17 studies, 14 were graded as high quality and 3 as moderate quality (S3 Table). Inter-author agreement in grading of studies was 88% (15/17). Overall agreement in quality assessment scores was 64% (11/17).

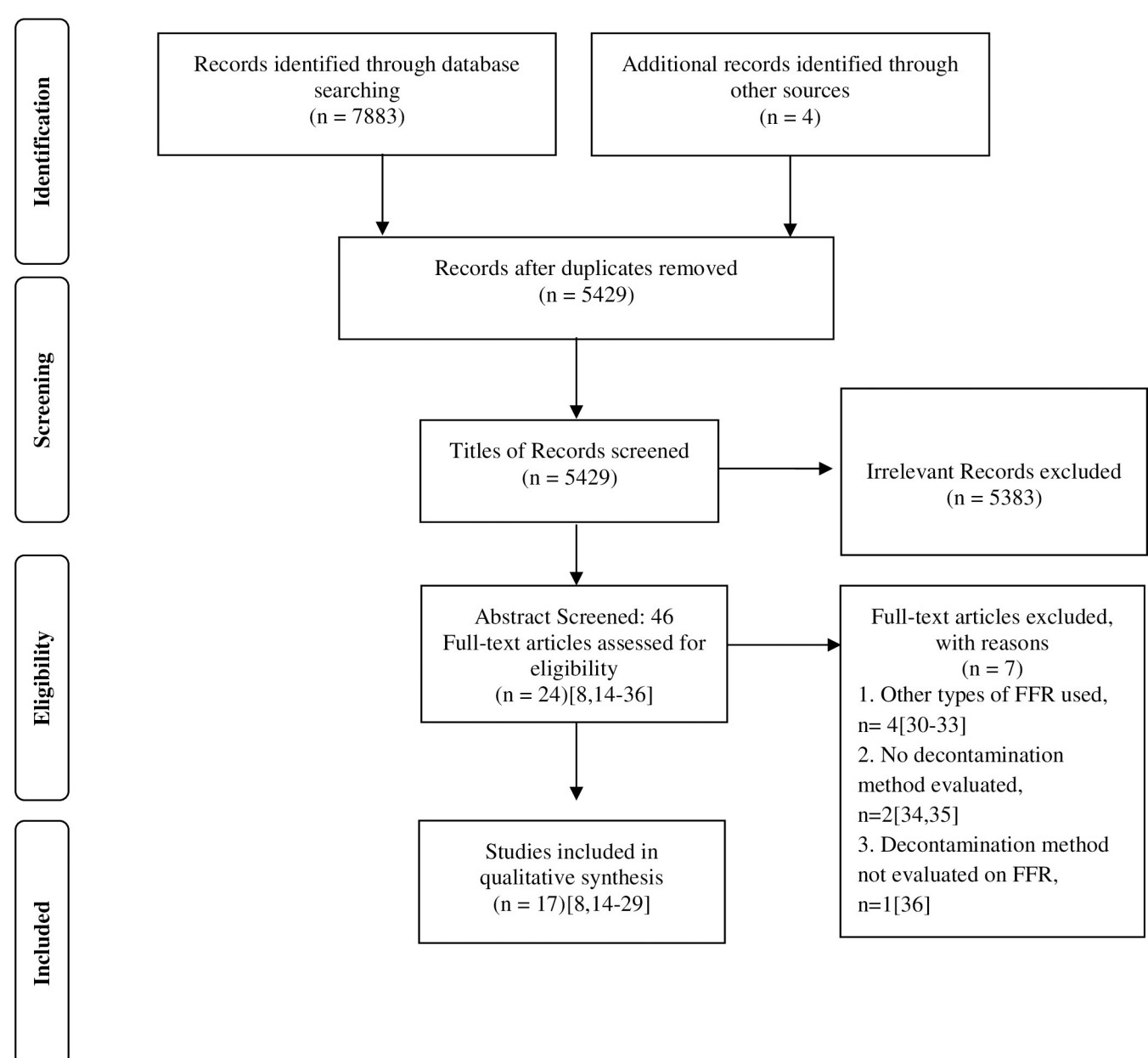

**Fig 1. Summary of search, selection and inclusion process.** Excluded studies [30–36] Abbreviations: FFR: Filtering Facepiece Respirator, n: Number.

## Study characteristics

Amongst 17 included studies, 15 were conducted in U.S. [8, 14–27] and 2 in Taiwan [28, 29]. Ten out of 15 studies were conducted by research groups from NIOSH as the principal investigator [8, 14, 16, 17, 21–26], 4 by researchers at Applied Research Associates (ARA) in collaboration with Air Force Research Laboratory at Tyndall Air Force Base, Panama City [18–20, 27] and in 1 study, principal investigators were from University of Nebraska (UoN) [15]. Three studies were an outcome of collaboration between NIOSH, ARA & UoN in various combinations [14, 15, 18]. Two studies from Taiwan were conducted by same researchers at Department of Occupational Safety and Health, Chung Shan Medical University [28, 29]. First study evaluating reprocessing methods for FFRs was published in 2007 [22] and last study in 2018 [29].

## Decontamination/reprocessing methods

Overall, 21 different types of decontamination or reprocessing methods for N95-FFRs were evaluated in included studies against various parameters namely physical changes, user acceptability, respirator fit, filter efficiency, microbicidal efficacy and presence of chemical residues post-reprocessing. Number of studies conducted for each reprocessing method, on these parameters are given in Fig 2. Overall, these studies evaluated 9 Physical (Energetic) reprocessing methods namely Ultraviolet (UV-C) Irradiation (UVGI) [8, 14–16, 19, 20, 23–25, 27, 29], UV-A [29], UV-B [27], Moist heat delivered using Microwave generated Steam (MGS) [14, 15, 20, 22, 23, 26], Lab Incubator (MHI) [14, 15, 20, 22, 23] and Autoclave (MHA) [22, 28, 29], Dry heat delivered by Microwave (MGI) [16, 22], Hot Air Oven (DHO) [22] and Traditional Electric Rice Cooker (TERC) [28, 29]; 3 Gaseous chemical decontamination methods namely Hydrogen Peroxide Gas Plasma (HPGP) [14, 16, 22, 27], Hydrogen Peroxide Vapor (HPV) [14] and Ethylene Oxide (EO) [14, 16, 22, 27]; 6 Liquid chemical decontamination methods namely Bleach [14, 16, 22, 25–29], Hydrogen Peroxide (LHP) [14, 22], Alcohols [22, 28, 29], Mixed Oxidants [27], Dimethyl dioxirane [27] and Soap & water [22]; and in one study [18], wipes of Bleach (0.9%), Benzalkonium chloride and Inert substance for surface decontamination of N95-FFRs. Fourteen (14) studies [14–18, 20–29] did comparative evaluation of multiple methods for reprocessing of FFRs whereas in 3 studies only 1 method was evaluated, which was UVGI in all [8, 19, 24]. In 12 studies [14–16, 16–23, 25, 27], intact respirators were exposed to the decontamination method whereas in 5, cut pieces of facepiece portion were exposed [8, 24, 26, 28, 29]. Furthermore, in one study [8], pieces of straps were also exposed separately to UVGI. In 4 studies, FFRs underwent multiple cycles (3 in all studies) of decontamination for reprocessing [14, 17, 18, 23].

## Respirator models

In 10 of 17 studies, the identities of N95-FFR models used was disclosed [8, 15, 17, 19, 21, 23–25, 28, 29], details of which against the reprocessing method and parameters evaluated are given in S4 Table. Overall, 23 different models of N95-FFRs were disclosed in 10 studies, 19 of which are approved as surgical respirators by FDA, whereas 4 are Particulate respirators. All respirators used in these studies, irrespective of whether identities were disclosed or not, were NIOSH approved. 3M1860 [8, 15, 19, 21, 23], 3M1870 [15, 17, 19, 21, 23] & 3M8210 [21, 23, 24, 28, 29] were the most commonly used N95-FFRs, each being used in 5 studies. 3M1860 & 3M1870, both surgical respirators were tested against three reprocessing methods i.e. UVGI, MGS and MHI, where identity was disclosed whereas 3M8210, a particulate respirator was exposed to 7 different reprocessing methods. Furthermore, in 2 studies, P100 respirators were also evaluated but in both identities were not disclosed [16, 22].

| PARAMETERS/ METHODS | PHYSICAL | USER ACCEPTABILITY* | RESPIRATOR FIT | FILTER EFFICIENCY | MICROBI-CIDAL EFFICACY | CHEMICAL RESIDUES |
|---|---|---|---|---|---|---|
| **PHYSICAL METHODS** | | | | | | |
| UVGI (TYPE-C) | 7[8,14,16,20-23] | 1[21] | 2[21,23] | 5[8,14-16,22] | 6[15,19,20,24,25,29] | 1[27] |
| UV-A | | | | | 1[29] | |
| UV-B | | | | | | 1[27] |
| MGS$^α$ | 4[14,20,21,23] | 1[21] | 2[21,23] | 3[14,15,17] | 4[15,17,20,26] | |
| MHI | 4[14,20,21,23] | 1[21] | 2[21,23] | 3[14,15,17] | 2[15,20] | |
| MHA | 1[22] | | | 2[22,28] | 1[29] | |
| MGI | 2[16,22] | | | 2[16,22] | | |
| DHO (TILL 80°C) | 1[22] | | | 2[16,22] | | |
| TERC | | | | 1[28] | 1[29] | |
| **GASEOUS CHEMICAL METHODS** | | | | | | |
| EO | 3[14,16,22] | | | 3[14,16,22] | | 1[27] |
| HPGP | 3[14,16,22] | | | 3[14,16,22] | | 1[27] |
| HPV | 1[14] | | | 1[14] | | |
| **LIQUID CHEMICAL METHODS** | | | | | | |
| LHP | 2[14,22] | | | 2[14,22] | | 1[27] |
| BLEACH | 3[14,16,22] | 3[14,16,27] | | 4[14,16,22,28] | 3[25,26,29] | 2[16,27] |
| ALCOHOLS$^β$ | 1[1] | | | 2[6,11] | 1[4] | |
| SOAP & WATER | 1[22] | | | 1[22] | | |
| MIXED OXIDANTS | | | | | | 1[27] |
| DIMETHYL DIOXIRANE | | | | | | 1[27] |
| **MISCELLANEOUS METHODS** | | | | | | |
| BLEACH WIPES | | | | 1[18] | 1[18] | |
| BAC WIPES | | | | 1[18] | 1[18] | |

**Fig 2. Summary of studies [Total Number, n[Reference] conducted, prior to 2020, on various parameters related to reprocessing of N95 Filtering Facepiece Respirators (FFRs).** Coloured cells represent cumulative results of these studies (See Legend Below). Numbers in each coloured cells represent total number of studies conducted on a reprocessing method: parameter combination. Numbers in Parentheses denote the reference number of studies. **Green Cells:** Evidence shows no

negative effect of the reprocessing method on the evaluated parameter. **Red Cells:** Evidence shows a negative effect of the reprocessing method on the evaluated parameter. **Orange Cells:** Evidence shows an effect which is either in conflict in different studies or requires careful consideration. **Grey Cells:** No study done on the reprocessing method: parameter combination. * User Acceptability is a composite parameter including odor, wear comfort & donning ease. References 14,16,27 only evaluated odor. α- Fisher *et al* 2011 [17] used Commercial steam bags for generation of steam, other studies used a water reservoir. β- Ethanol (70%) [28, 29] and Isopropyl alcohol (70% [28] and 100% [22]) were used. Abbreviations: **UVGI:** Ultraviolet Irradiation (Type-C, 254 nm), **MGS:** Microwave Generated Steam, **MHI:** Moist heat Incubation in Lab Incubator, **MHA:** Moist Heat in Autoclave, **DHO:** Dry Heat in Oven (Till 80˚C), **TERC:** Traditional Electric Rice Cooker, **EO:** Ethylene Oxide, **HPGP:** Hydrogen Peroxide Gas Plasma, **HPV:** Hydrogen Peroxide Vapor, **LHP:** Liquid Hydrogen Peroxide, **BAC:** Benzalkonium Chloride. Note: The summary is only indicative of the collective results of various studies done (prior to 2020) to evaluate effect of reprocessing method on a particular parameter. It doesn't attempt to endorse or refute any method as the authors strongly believe that there is insufficient data to reach any conclusion.

## Decontamination methods

**A. Physical (Energetic) methods.** *i. Ultra-Violet Irradiation (UVGI).* Thirteen studies [8, 14–16, 19–25, 27, 29] evaluated exposure to UV-C (254 nm) as a reprocessing method for FFRs, as shown in Fig 2. All 23 known models of N95-FFRs were reprocessed using UV-C in at least one study (S4 Table). Furthermore, one study each also examined the microbiological efficacy of UV-A [29] and presence of chemical residues after using UV-B [27]. Exposure variables of UVGI (UV-C) on N95-FFRs and summary of results are provided in Table 1. Different parameters evaluated against UVGI are detailed in Fig 2. Overall, UVGI has shown to be microbiologically efficacious [15, 19, 20, 24, 25, 29], preserve physical appearance of FFRs [8, 14, 16, 20–23] & their filter efficiency [8, 14–16, 22], acceptable to users in terms of odor, donning ease and wear comfort [21], maintain respirator fit [21, 23] and devoid of any toxic residues post-exposure [27]. UVGI has shown to preserve filter efficiency & achieve adequate microbicidal efficacy post-exposure in 9 [8, 15, 24] & 18 different N95-FFR models [15, 19, 24, 25, 29], respectively, where identity of models was disclosed.

*ii. Moist heat.* Delivering moist heat to FFRs has been evaluated in 10 studies [14, 15, 17, 18, 21–23, 26, 28, 29]. Modalities of exposure involved exposing FFRs to steam created in a microwave (MGS), either by using water reservoir [14, 15, 20, 21, 23, 26] or commercial steam bags [17]; in a lab incubator with a water reservoir heated at 60-70˚C (MHI) [14, 15, 18, 21, 23] and by autoclaving at 121˚C (MHA) [22, 28, 29]. Parameters evaluated for these treatments are given in Fig 2 and the exposure variables and results of individual studies are described in Table 2. Known FFR models which underwent reprocessing by both MGS and MHI were 3M1860, 3M1870, 3M8000 and 3M8210, whereas, for MHA only known FFR model was 3M8210. MHA physically destroyed FFRs thus deemed unsuitable for further evaluation [22]. Both MGS & MHI methods showed acceptable microbiological efficacy [15, 17, 20, 26] and no significant effect on user acceptability [21], respirator fit [21, 23] and filter efficiency [14, 15, 17], till 3 cycles of decontamination.

*iii. Dry heat.* Dry heat for reprocessing of FFRs has been evaluated in 4 studies [16, 22, 28, 29] wherein microwave (MGI) [16, 22], Hot Air Oven (DHO) [16, 22] and Electric Rice Cooker (TERC) [28, 29] have been used. 3M8210 was the only known N95-FFR model which underwent reprocessing by any dry heat delivering method [28, 29]. Various parameters which have been evaluated against them are shown in Fig 2 and their exposure variables and results are summarized in Table 2. In MGI method, respirator models were destroyed in both studies [16, 22]. FFRs reprocessed by DHO were able to physically withstand temperatures at 80˚C without affecting durability and filter efficiency [16, 22]. Electric rice cooker (TERC) was able to provide 99–100% biocidal efficacy against *Bacillus subtilis* spores [29].

**B. Gaseous chemical methods.** Only 4 studies [14, 16, 22, 27], prior to 2020, had evaluated a gaseous disinfection method for reprocessing of N95-FFRs. The methods used were Ethylene Oxide (EO) [14, 16, 22, 27], Hydrogen peroxide in a Plasma Sterilizer (HPGP) [14, 16, 22, 27] and Hydrogen Peroxide in vaporized form by using a commercial automated vapor

**Table 1. Summary of characteristics of studies using Ultraviolet Irradiation (UVGI) as a reprocessing method for N95-FFRs.**

| Authors (Year) | Type | Variables of UVGI Irradiation | | | | | Variables of FFRs | | | Results | |
|---|---|---|---|---|---|---|---|---|---|---|---|
| | | Irradiance (mW/cm$^2$) | Duration | Dose (J/cm$^2$) | Sides Exposed to UVGI | No. of Cycle | Total no. of Models used | Part of FFR exposed to UVGI | Repli-cates | Parameters Assessed | Summary of Results |
| Bergman et al [14] (2010) | C | 1.8 | 45 m | - | Outer (Convex) | 3 | 6 | Intact | 3 | Physical Changes | No observable physical changes on FFRs |
| | | | | | | | | | | Odor | No comment on odor |
| | | | | | | | | | | Filter Efficiency | Expected levels of Filter Aerosol penetration (<5%) & filter airflow resistance |
| Lore et al [15] (2012) | C | 1.6–2.2 | 15 m | 1.8 | Outer (Convex) | 1 | 2 | Intact | 9 | Filter Efficiency | No significant degradation of filter performance |
| | | | | | | | | | | Microbicidal Efficacy | >4 log$^{10}$ TCID$_{50}$/ml reduction of H5N1 Avian Influenza virus |
| Viscusi et al [16] (2009) | C | 0.18–0.2 | 30 m | 0.17–0.18 | Each side | 1 | 9 | Intact | 3 | Physical Changes | No observable physical changes on FFRs |
| | | | | | | | | | | Filter Efficiency | Didn't affect Filter efficiency |
| Lindsley et al [8] (2015) | C | | | 120, 240, 470, 950 (For mask layers); | NA | 1 | 4 | Facepiece Coupons and Straps | 4 | Structural Integrity | Strengths of respirator materials was substantially reduced (in some cases>90%) |
| | | | | 590, 1180, 2360 (For straps, each side) | | | | | | Filter Efficiency | Slight increase in particle penetration but no effect on airflow resistance |
| Mills et al [19] (2018) | C | 17 | 60–70 s | 1 | Outer (Convex) | 1 | 15 | Intact | 3 | Microbicidal Efficacy | ≥3 log$_{10}$ TCID$_{50}$/ml reduction in Influenza virus (H1N1) viability on 12/15 FFR models and straps from 7/15 FFR models |
| Heimbuch et al [20] (2011) | C | 1.6–2.2 | 15 m | 1.8 | Outer (Convex) | 1 | 6 | Intact | 3 | Physical Changes | No observable physical changes on FFRs |
| | | | | | | | | | | Microbicidal Efficacy | >4 log$_{10}$ TCID$_{50}$/ml reduction of Influenza virus (H1N1) |
| Viscusi et al [21] (2011) | C | 1.8 | 30 m | - | Each side | 3 | 6 | Intact | 2 | Physical Changes | No observable physical changes on FFR |
| | | | | | | | | | | User Acceptability | No clinically meaningful reduction in respirator fit, increase in odor, increase in discomfort or increased difficulty in donning |
| | | | | | | | | | | Respirator Fit | |
| Viscusi et al [22] (2007) | C | 0.24 | 15/ 240 m | - | Each side | 1 | 2 | Intact | 4 | Physical Changes | No observable physical changes on FFRs |
| | | | | | | | | | | Filter Efficiency | Not significantly affected by both time durations on both types of FFRs (N95 and P100) |

(*Continued*)

**Table 1.** (Continued)

| Authors (Year) | Type | Variables of UVGI Irradiation | | | | | Variables of FFRs | | | Results | |
|---|---|---|---|---|---|---|---|---|---|---|---|
| | | Irradiance (mW/cm²) | Duration | Dose (J/cm²) | Sides Exposed to UVGI | No. of Cycle | Total no. of Models used | Part of FFR exposed to UVGI | Repli-cates | Parameters Assessed | Summary of Results |
| Bergman *et al* [23] (2011) | C | 1.8 | 15 m | - | Outer (Convex) | 3 | 3 | Intact | 2 | Physical Changes | No observable physical changes on FFRs |
| | | | | | | | | | | Respirator Fit | No significant changes in Respirator fit |
| Fisher *et al* [24] (2010) | C | 2.5 | 1, 2, 4, 10 m on 3M 8210,1870 | 0.03, 0.1 & 0.3 on Wilson, 3M 1860 and KC | Each side | 1 | 6 | Facepiece Coupons | 3 | IFM specific dose for | Log Reduction of MS2 Coliphage is a function of FFR model specific IFM UV-C dose |
| | | | 10m on Cardinal N95-ML | | | | | | | Microbicidal Efficacy | |
| Lin *et al* [29] (2018) | C | 18.9 | 1, 2, 5, 10, 20 m | - | NA | 1 | 1 | Cut pieces | 3 | Microbicidal Efficacy | 99–100% biocidal efficacy against *Bacillus subtilis* spores |
| Vo *et al* [25] (2009) | C | 0.4 | 1, 2, 3, 4, 5 hr | 1.44, 2.88, 4.32, 5.76, 7.2 | One side | 1 | 1 | Intact | 3 | Microbicidal Efficacy | 3 log reduction of MS2 Coliphage at dose of 4.32 J/cm² and complete removal at dose of ≥7.2 J/cm² |
| Salter *et al* [27] (2010) | C | 3.4 | 1 hr | 27 | NA | 1 | 6 | Coupons, straps, | 3 | Presence of Toxic Chemical residues Post-exposure | No toxic residues post-exposure |
| | | | | | | | | Nose cushion, | | | |
| | | | | | | | | Nose pieces | | | |
| Lin *et al* [29] (2018) | A | 31.2 | 1, 2,5, 10, 20 m | - | Each side | 1 | 1 | Cut pieces | 3 | Microbicidal Efficacy | Poor Microbicidal efficacy against *Bacillus subtilis* spores |
| Salter *et al* [27] (2010) | B | 4 | 1 hr | - | NA | 1 | 6 | Coupons, straps, nose cushion, | 3 | Presence of Toxic Chemical residues Post-exposure | No toxic residues post-exposure |
| | | | | | | | | Nose pieces | | | |

**ABBREVIATIONS: mW/cm²**: milli Watt per square centimetre, **J/cm²**: Joules per square centimetre **m**: Minute, **NA**: Not Applicable, **FFR**: Filtering Facepiece Respirator, **TCID**: Tissue Culture Infectious Dose, **s**: Seconds **IFM**: Internal Filtering Media, **hr**: Hour

generator [22]. FFR models were not disclosed in any of the studies. Parameters against which they were evaluated; and their exposure variables and findings of the studies are provided in Fig 2 and Table 3, respectively. After EO sterilization, FFRs didn't showed any physical changes [14, 16, 22], or had offensive odor [14, 16], and filter efficiency was also not degraded significantly [14, 16, 22] even after undergoing 3 cycles [14]. In 3 studies, where HPGP was evaluated, no significant physical changes on the FFRs were noted [14, 16, 22] but filter efficiency of 25% (9/36) respirators was noted to be degraded in one [14] of three [14, 16, 22] studies. However, similar effect was not noted when FFRs were treated with vaporized form [14, 22].

**C. Liquid chemical methods.** Six different liquid decontamination methods have been evaluated on N95-FFRs in 8 studies [14, 16, 22, 25–29]. These are Bleach [14, 16, 22, 25–29], Liquid Hydrogen Peroxide (LHP) [14, 22, 27], Alcohols [22, 28, 29] including Ethanol and Iso-propyl Alcohol, Mixed oxidants [27], Dimethyl Dioxirane [27] and Soap solution [22]. Param-eters against which they were evaluated, their exposure variables and results of the studies are provided in Fig 2 and Table 4, respectively. Against Bleach, only known N95-FFR models

**Table 2. Summary of characteristics of studies using physical decontamination methods, other than UVGI, for reprocessing of FFRs.**

| Authors (Year) | Variables of Decontamination Methods | | | | Variables of FFRs | | | Results | |
| | Mode of Delivery | Temperature | Duration | No. of Deconta-mination Cycle | Total no. of Models used | Part of FFR exposed | Replicates | Parameters Assessed | Summary of Results |
|---|---|---|---|---|---|---|---|---|---|
| **DRY HEAT** | | | | | | | | | |
| Viscusi et al [16] (2009) | Microwave | - | 2 m (1 m each side) | 1 | 9 (6 N95 3 P100) | Intact | 3 | Physical Changes | Observable physical changes on many models of FFRs |
| | | | | | | | | Filter Efficiency | Expected levels of Filter Aerosol penetration (<5%) & filter airflow resistance |
| Viscusi et al [22] (2007) | Microwave | - | 2 and 4 m (1 & 2 m each side) | 1 | 2 (1 N95 1 P100) | Intact | 4 | Physical Changes | No visible changes after 2 min for both models |
| | | | | | | | | | Visible damage after 4 min for both models |
| | | | | | | | | Filter Efficiency | Filter efficiency not significantly changed after 2 min for both models |
| | | | | | | | | | Filter efficiency of N95-FFR was significantly increased after 4 min |
| Viscusi et al [16] (2009) | Hot air Oven | 80-120° C | 1 hr | 1 | 9 (6 N95 3 P100) | Intact | 3 | Physical Changes | No Comment |
| | | | | | | | | Filter Efficiency | Temperature affected filter aerosol penetration and component melting which was model specific |
| Viscusi et al [22] (2007) | Hot air oven | 80° C & 160° C | 1 hr | 1 | 2 (1 N95 1 P100) | Intact | 4 | Physical Changes | No visible changes for either type of respirator at 80° C |
| | | | | | | | | Filter Efficiency | Complete destruction of both types of respirators at 160° C |
| | | | | | | | | | Small increase in average penetration for both types of respirators |
| Lin et al [28] (2017) | Rice Cooker | 149-164° C | 3 m | 1 | 1 | Cut pieces | 3 | Filter Efficiency | Decontamination reduced the filter quality but less than liquid chemical methods |
| Lin et al [29] (2018) | Rice Cooker | 149-164° C | 3 m | 1 | 1 | Cut pieces of FFR layers | 3 | Microbicidal Efficacy | 99–100% Biocidal efficacy against *Bacillus subtilis* spores |
| **MOIST HEAT** | | | | | | | | | |
| Bergman et al [14] (2010) | Microwave (MGS) | | 2 m | 3 | 6 | Intact | 3 | Physical Changes | Partial separation of inner foam cushion of 1 FFR model |
| | | | | | | | | Odor | No comment on odor |
| | | | | | | | | Filter Efficiency | Expected levels of filter aerosol penetration (<5%) & filter airflow resistance |
| Lore et al [15] (2012) | Microwave (MGS) | | 2 m | 1 | 2 | Intact | 9 | Filter Efficiency | No significant degradation of filter performance |
| | | | | | | | | Microbicidal Efficacy | >4 $\log_{10}$ TCID$_{50}$/ml reduction of H5N1 Avian Influenza virus |
| Fisher et al [17] (2011) | Microwave (MGS) | | 90 s | 3 | 3 | Intact | 3 | Microbicidal Efficacy | >3 $\log_{10}$ reduction in pfu/FFR of MS2 Coliphage |

*(Continued)*

**Table 2.** (Continued)

| Authors (Year) | Mode of Delivery | Temperature | Duration | No. of Deconta-mination Cycle | Total no. of Models used | Part of FFR exposed | Replicates | Parameters Assessed | Summary of Results |
|---|---|---|---|---|---|---|---|---|---|
| | **Variables of Decontamination Methods** | | | | **Variables of FFRs** | | | **Results** | |
| Heimbuch et al [20] (2011) | Microwave (MGS) | | 2 m | 1 | 6 | Intact | 3 | Physical Changes | Slight separation of foam nose cushion in 1 FFR model |
| | | | | | | | | Microbicidal Efficacy | >4 $\log_{10}$ $TCID^{50}$/ml reduction of Influenza virus (H1N1) |
| Viscusi et al [21] (2011) | Microwave (MGS) | | 2 m | 1 | 6 | Intact | 2 | Physical Changes | Slight separation of inner foam nose cushion in 1 FFR model |
| | | | | | | | | User Acceptability | No significant changes in odor, increase in discomfort or increased difficulty in donning |
| | | | | | | | | | Strap breakage during multiple donning not more frequent than in controls |
| | | | | | | | | Respirator Fit | No clinically meaningful reduction in respirator fit |
| Bergman et al [23] (2011) | Microwave (MGS) | | 2 m | 3 | 3 | Intact | 2 | Physical Changes | Slight separation of inner foam nose cushion in 1 FFR model |
| | | | | | | | | Respirator Fit | No significant changes in Respirator fit |
| Fisher et al [26] (2009) | Microwave (MGS) | | 15, 30, 45, 60, 75, 90 s | 1 | 1 | Cut pieces | 4 | Microbicidal Efficacy | >4 $\log_{10}$ reduction in MS2 Coliphage pfu/ml after $\geq$ 45 seconds |
| Bergman et al [14] (2010) | Lab Incubator (MHI) | 60°C | 30 m | 3 | 6 | Intact | 3 | Physical Changes | Partial separation of inner foam cushion of 1 FFR model |
| | | | | | | | | Odor | No comment on odor |
| | | | | | | | | Filter Efficiency | Expected levels of Filter Aerosol penetration (<5%) & filter airflow resistance |
| Lore et al [15] (2012) | Lab Incubator (MHI) | 65 ± 5°C | 3 hr | 1 | 2 | Intact | 9 | Filter Efficiency | No profound reduction in filter efficiency |
| | | | | | | | | Microbicidal Efficacy | >4 $\log_{10}$ $TCID_{50}$/ml reduction of H5N1 Avian Influenza virus achieved |
| Heimbuch et al [20] (2011) | Lab Incubator (MHI) | 65 ± 5°C | 30 m | 1 | 6 | Intact | 3 | Physical Changes | No obvious signs of deformation or deterioration of FFRs |
| | | | | | | | | Microbicidal Efficacy | >4 $\log_{10}$ $TCID_{50}$/ml reduction of Influenza virus (H1N1) |
| Viscusi et al [21] (2011) | Lab Incubator (MHI) | 60°C | 30 m | 1 | 6 | Intact | 2 | Physical Changes | Slight separation of inner foam nose cushion in 1 FFR model |
| | | | | | | | | User Acceptability | Mean Odor scores were increased only for 1 FFR model |
| | | | | | | | | | No significant increase in discomfort or increased difficulty in donning |
| | | | | | | | | | Strap breakage during multiple donning not more frequent than in controls |
| | | | | | | | | Respirator Fit | No clinically meaningful reduction in respirator fit |

(*Continued*)

**Table 2.** (Continued)

| Authors (Year) | Variables of Decontamination Methods | | | | Variables of FFRs | | | Results | |
|---|---|---|---|---|---|---|---|---|---|
| | Mode of Delivery | Temperature | Duration | No. of Decontamination Cycle | Total no. of Models used | Part of FFR exposed | Replicates | Parameters Assessed | Summary of Results |
| Bergman *et al* [23] (2011) | Lab Incubator (MHI) | 60˚C | 15 m | 3 | 3 | Intact | 2 | Physical Changes | Slight separation of inner foam nose cushion in 1 FFR model |
| | | | | | | | | Respirator Fit | No significant changes in Respirator fit |
| Viscusi *et al* [22] (2007) | Autoclave (MHA) | 121˚C | 15/ 30 m | 1 | 2 | Intact | 4 | Physical Changes | N95-FFRs were deformed in both conditions and P100 FFRs were unchanged but respirator media felt softer |
| | | | | | | | | Filter Efficiency | Degradation in filter efficiency of both Respirator types |
| Lin *et al* [28] (2017) | Autoclave (MHA) | 121˚C | 15 m | 1 | 1 | Cut pieces of FFR facepiece | 3 | Filter Efficiency | Decontamination reduced the filter quality but less than liquid chemical methods |
| Lin *et al* [29] (2018) | Autoclave (MHA) | 149-164˚ C | 3 m | 1 | 1 | Cut pieces of FFR facepiece | 3 | Microbicidal Efficacy | 99–100% Biocidal efficacy against *Bacillus subtilis* spores |

**ABBREVIATIONS: UVGI:** Ultraviolet Irradiation, **FFR:** Filtering Facepiece Respirator, **m:** minute, **hr:** hour, **TCID:** Tissue Culture Infectious Dose, **s:** second, **pfu:** Plaque Forming Unit

evaluated were 3M8210 and Wilson SAF-T-FIT Plus (S4 Table). 3M8210 was the only known N95-FFR which was evaluated for Alcohols [28, 29].

**D. Miscellaneous methods. In one study [18], commercial wipes of 0.9% Sodium Hypochlorite, Benzalkonium Chloride and an Inert material were evaluated for changes in filter efficiency and microbicidal efficacy by applying them on surface of N95-FFRs, as shown in Fig 2 & Table 4.**

## Discussion

An Influenza pandemic was always on the horizon and in 2009, it became reality. Researchers at NIOSH have been looking actively for finding a suitable method for reprocessing of FFRs since 2006 after the report of IOM Committee to tackle global shortage of FFRs [11, 22]. Consequently, search for a suitable reprocessing method began under NIOSH. During 2007–2012, 12 studies were published which evaluated reprocessing methods for FFRs, most of them were conducted by or in collaboration with NIOSH [14–17, 20–27]. In contrast, between 2013–2019, only 5 published studies had evaluated a reprocessing technique for N95-FFRs [19, 20, 28, 29], with last study published by NIOSH in 2015 [8]. Ongoing COVID-19 pandemic has brutally exposed the stalled progress in research to address this issue.

It has been shown that the surface stability of SARS-CoV-2 on various surfaces lasts up to 3 days but this study didn't include porous surfaces like that of respirators [37]. However, a study recently, showed it to be present on outer layer of surgical masks on day 7 [38]. This recent data makes it imperative to decontaminate FFRs in between use as the risk of contact transmission without decontamination is considerable. Previously, CDC also discouraged reusing N95-FFRs whenever risk of contact transmission of a pathogen was high [6]. Furthermore, it is in larger global interest to find a suitable reprocessing method for N95-FFRs as they are not used frequently by HCWs in low to middle income countries (LMICs) while tackling

**Table 3. Summary of characteristics of studies using gaseous chemical methods for reprocessing of FFRs.**

| Authors | Variables of Decontamination Methods | | | | Variables of FFRs | | | Results | |
|---|---|---|---|---|---|---|---|---|---|
| | Disinfectant Sterilizer | Packaging Conditions | Duration | No. of Decontamination Cycles | Total no. of Models used | Part of FFR exposed | Replicates | Parameters Assessed | Summary of Results |
| Bergman et al [14] (2010) | Ethylene Oxide<br><br>Amsco® Eagle® 3017 | Kept in Tyvek® pouches<br><br>6 FFR per pouch | 1 hr exposure<br><br>12 hr aeration | 3 | 6 | Intact | 3 | Physical Changes | Partial separation of inner foam cushion of 1 FFR model |
| | | | | | | | | Odor | No comment on odor |
| | | | | | | | | Filter Efficiency | Expected levels of filter aerosol penetration (<5%) & filter airflow resistance |
| Viscusi et al [16] (2009) | Ethylene Oxide<br><br>3 M Steri-Vac 5XL | Individual poly/paper pouch | 1 hr exposure<br><br>4 hr aeration | 1 | 9<br><br>(6 N95<br><br>3 P100) | Intact | 3 | Physical Changes | No observable physical changes on FFRs |
| | | | | | | | | Filter Efficiency | Expected levels of filter aerosol penetration (<5%) & filter airflow resistance |
| Viscusi et al [22] (2007) | Ethylene Oxide<br><br>3 M Steri-Vac 4XL & 5 XL | Individual poly/paper pouch | 1 hr exposure<br><br>4 hr aeration | 1 | 2 | Intact | 4 | Physical Changes | Straps of P100 FFRs were slightly darkened |
| | | | | | | | | Filter Efficiency | Average penetration increased for both respirator types but were within NIOSH certification criteria |
| Salter et al [27] (2010) | Ethylene Oxide<br><br>Amsco® Eagle® 3017 | Individual sterilization pouch | 3 hr exposure<br><br>12 hr aeration | 1 | 6 | Intact | 3 | Presence of Toxic Chemical Residues | EO was not detected on any of the model |
| | | | | | | | | | Treated EO contained Diacetone alcohol and a possible mutagen and carcinogen, 2-hydroxyethyl acetate (HEA) |
| Bergman et al [14] (2010) | $H_2O_2$ Gas Plasma (HPGP)<br><br>STERRAD® 100S | Mylar/Tyvek® pouch<br><br>6 samples per pouch | 55 m cycle time | 3 | 6 | Intact | 3 | Physical Changes | No physical changes on FFRs |
| | | | | | | | | Odor | No comment on odor |
| | | | | | | | | Filter Efficiency | 25% (9/36) samples had aerosol penetration >5% suggestive of degradation in filter efficiency |
| Viscusi et al [16] (2009) | $H_2O_2$ Gas Plasma (HPGP)<br><br>STERRAD® 100S | Mylar/Tyvek® pouch<br><br>6 samples per pouch | 55 m cycle time | 1 | 9<br><br>(6 N95<br><br>3 P100) | Intact | 3 | Physical Changes | Metallic nose bands not as shiny as unexposed controls |
| | | | | | | | | Filter Efficiency | Expected levels of Filter Aerosol penetration (<5%) & filter airflow resistance |
| Viscusi et al [22] (2007) | $H_2O_2$ Gas Plasma (HPGP)<br><br>STERRAD® 100S<br><br>STERRAD® NX | Mylar/Tyvek® pouch | <br><br>55 m<br><br>100 m | 1 | 2 | Intact | 4 | Physical Changes | Aluminium nosebands slightly tarnished with both cycles |
| | | | | | | | | Filter Efficiency | Average penetration not significantly increased & remained within limit of NIOSH certification criteria for both respirator types and cycling conditions |
| Salter et al [27] (2010) | $H_2O_2$ Gas Plasma (HPGP)<br><br>STERRAD® 100S | Sterilization pouches | 55 m | 1 | 6 | Intact | 3 | Presence of Toxic Chemical Residues | No residues on FFRs |
| | | | | | | | | | Sterilization cycle aborted when >6 FFRs were loaded in the sterilization chamber |

(Continued)

**Table 3.** (Continued)

| Authors | Variables of Decontamination Methods | | | | Variables of FFRs | | | Results | |
|---|---|---|---|---|---|---|---|---|---|
| | Disinfectant Sterilizer | Packaging Conditions | Duration | No. of Decontamination Cycles | Total no. of Models used | Part of FFR exposed | Replicates | Parameters Assessed | Summary of Results |
| Bergman *et al* [14] (2010) | $H_2O_2$ Vapor (HPV) | | 15 m dwell | 3 | 6 | Intact | 3 | Physical Changes | No physical changes on FFRs |
| | | | | | | | | Odor | No comment on odor |
| | Clarus® R HPV Generator | | 125 m total cycle time | | | | | Filter Efficiency | Expected levels of filter aerosol penetration (<5%) & filter airflow resistance |

**ABBREVIATIONS: FFR:** Filtering Facepiece Respirator, **hr:** Hour, **m:** Minute, **$H_2O_2$**: Hydrogen Peroxide

airborne pathogens, such as *Mycobacterium tuberculosis*, against which their use is mandatory [39, 40]. Finding a reprocessing method for FFRs will lead to provision of adequate respiratory protection for HCWs in such resource limited settings.

We found that UVGI was the most frequently evaluated reprocessing method for N95-FFRs, as shown in Fig 2 and Table 3. Reprocessing by UVGI method maintained the overall physical structure and filter efficiency of the FFRs and was able to demonstrate sufficient microbicidal efficacy. Furthermore, it had insignificant influence on the respirator fit and reprocessed FFRs were devoid of any toxic residues but studies which evaluated these parameters were few. Furthermore, these findings should be assessed in view of varying exposure variables of UV dose used in these studies and the methodological variations in estimating the measures of microbicidal efficacy, as shown in Table 1.

Dose of irradiation is the most important variable for determining microbiological efficacy of UVGI method which, in turn, is determined by irradiance at the surface of FFR and duration of exposure [19]. All studies [15, 19, 20, 24, 25], except one [29], which evaluated the microbicidal efficacy of UVGI used enveloped viruses as the challenge micro-organism. Total doses around 1–2 J/cm$^2$ have shown to provide $\geq$4 log$_{10}$ reduction of viruses inoculated on FFRs [15, 20, 25]. Lin *et al* [29] used *Bacillus subtilis* spores as challenge micro-organisms and found that from a 18.9mW/cm$^2$ UV-C source, exposure for 5 min (corresponding to a dose of 5–6 J/cm$^2$) was able to kill all spores. However, this study measured relative survival of spores (in percentage) on exposed respirator coupons as compared to control coupons (unexposed) instead of log reduction of spores. Whether such doses will be effective against other airborne pathogens, such as *M. tuberculosis* should be assessed in future research. Furthermore, a study by Fisher *et al* [24] concluded that the UV-C dose required for microbicidal efficacy is a function of the dose available to the electret medium rather than total dose, which in turn, is dependent on the penetrance (transmittance) of the layer above it. Hence, effective doses of UV-C for microbicidal efficacy will be model specific and needs to be established accordingly. We conclude that UVGI has great potential to be utilized as an effective decontamination method for N95-FFRs during this time of crisis however, more studies are needed to validate the various variables associated with the delivery of the UVGI method and respirator model specific doses will need to be established.

MGS & MHI methods delivered moist heat to FFRs in a microwave and a bench top laboratory incubator, respectively and have shown no significant effect on user acceptability, respirator fit and filter efficiency till 3 cycles of decontamination [14, 15, 21, 23]. However, multiple studies evaluating physical changes noticed partial separation of inner foam nose cushion in both methods for a particular FFR model (3M1870), where model identity was disclosed, but

**Table 4. Summary of characteristics of studies using liquid & miscellaneous chemical methods for reprocessing of FFRs.**

| Authors | Variables of Decontamination Methods | | | | Variables of FFRs | | | Results | |
|---------|------------|---------------|----------|----------------------------------|--------------------------------|---------------------|------------|------------------------|-----------------------------|
| | Disinfectant | Concentration | Duration | No. of Decontamination Cycles | Total no. of Models used | Part of FFR exposed | Replicates | Parameters Assessed | Summary of Results |
| Bergman *et al* [14] (2010) | Liquid $H_2O_2$ (LHP) | 6% | 30 m Submersion | 3 | 6 | Intact | 3 | Physical Changes | Staples were oxidized to varying degree |
| | | | | | | | | Odor | No comment on odor |
| | | | | | | | | Filter Efficiency | Expected levels of Filter Aerosol penetration (<5%) & filter airflow resistance |
| Viscusi *et al* [22] (2007) | Liquid $H_2O_2$ (LHP) | 3% | 30 m submersion | 1 | 2 (1 N95 1 P100) | Intact | 4 | Physical Changes | No observable changes on both respirator types with 3% $H_2O_2$ & slight fading of label ink with 6% $H_2O_2$ |
| | | 6% | | | | | | Filter Efficiency | Average penetration within NIOSH certification limit for both respirator types & both concentrations |
| Salter *et al* [27] (2007) | Liquid $H_2O_2$ (LHP) | 3% | 30 m submersion | 1 | 6 | Intact | 3 | Presence of Toxic Chemical Residues | No deposition of significant quantities of toxic residues on FFRs |
| Bergman *et al* [14] (2010) | NaOCl (Bleach) | 0.6% | 30 m Submersion | 3 | 6 | Intact | 3 | Physical Changes | Metallic nosebands were tarnished, Staples were oxidized to varying degree, discoloured inner nose pads, dry to touch |
| | | | | | | | | Odor | All FFRs had a characteristic bleach odor after overnight air drying |
| | | | | | | | | Filter Efficiency | Expected levels of filter aerosol penetration (<5%) & filter airflow resistance |
| Viscusi *et al* [16] (2009) | NaOCl (Bleach) | 0.6% | 30 m Submersion | 1 | 9 | Intact | 3 | Physical Changes | Metallic nose bands were tarnished |
| | | | | | | | | Odor | All FFRs had a scent of bleach and after rehydration with water, increase in chlorine off-gassing was measured |
| | | | | | | | | Filter Efficiency | Expected levels of filter aerosol penetration (<5%) & filter airflow resistance |
| Lin *et al* [28] (2017) | NaOCl (Bleach) | 0.5% | 10 m Submersion | 1 | 1 | Cut pieces of facepiece | 3 | Filter Efficiency | Decontamination reduced the filter quality |

(*Continued*)

**Table 4.** (*Continued*)

| Authors | Variables of Decontamination Methods | | | | Variables of FFRs | | | Results | |
|---------|------------|---------------|----------|------------------------------------|------------------------------|------------------------|------------|-----------------------|-------------------|
| | Disinfectant | Concentration | Duration | No. of Decontamination Cycles | Total no. of Models used | Part of FFR exposed | Replicates | Parameters Assessed | Summary of Results |
| Viscusi *et al* [22] (2007) | NaOCl (Bleach) | 0.52%<br><br>5.2% | 30 m Submersion (both) | 1 | 2<br>(1 N95<br>1 P100) | Intact | 4 | Physical Changes | Aluminium nose bands were tarnished at both concentrations |
| | | | | | | | | Filter Efficiency | At 0.52% & 5.2% conc., average penetration for both respirator types were within NIOSH certification criteria |
| Lin *et al* [29] (2018) | NaOCl (Bleach) | 0.54%<br>2.7%<br>5.4% | NA Inoculated | 1 | 1 | Cut pieces of Face-piece | 3 | Microbicidal Efficacy | 100% Biocidal efficacy against *Bacillus subtilis* spores at the lowest concentration |
| Vo *et al* [25] (2009) | NaOCl (Bleach) | 0.005/0.01/0.05/0.1/<br>0.25/0.5/<br>0.75% | 10 m Submersion | 1 | 1 | Intact | 3 | Microbicidal Efficacy | $\geq$0.5% bleach causes 4 $\log_{10}$ reduction in pfu/ml of MS2 Coliphage |
| Fisher *et al* [26] (2009) | NaOCl (Bleach) | 0.0006%, 0.006%, 0.06%, 0.6% | 2 m Submersion | 1 | 1 | Cut Coupons of Face-piece | 3 | Microbicidal Efficacy | 0.6% bleach causes 4 $\log_{10}$ reduction in pfu/ml of MS2 Coliphage |
| Salter *et al* [27] (2010) | NaOCl (Bleach) | 0.6% | 30 m Submersion | 1 | 6 | Intact | 3 | Physical changes | Corrosion of metal parts was noted |
| | | | | | | | | Odor | FFRs retained a bleach odor following an off-gas period of 18 hour |
| | | | | | | | | Presence of Toxic Chemical Residues | Measured amount of residual chlorine was below permissible exposure limit |
| Viscusi *et al* [22] (2007) | Soap & Water | 1g/L | 2 m<br>20 m Submersion (both) | 1 | 2<br>(1 N95<br>1 P100) | Intact | 4 | Physical Changes | No physical changes observed for both durations |
| | | | | | | | | Filter Efficiency | Average penetration increased for both durations and both respirators |
| Salter *et al* [27] (2007) | Mixed Oxidants | (10% Oxone, 6% Sodium Chloride, 5% Sodium Bicarbonate) | 30 m submersion | 1 | 6 | Intact | 3 | Physical Changes | Oxidised metal parts |
| | | | | | | | | Odor | Left distinct odor on FFRs |
| | | | | | | | | Presence of Toxic Chemical Residues | No comment |
| Salter *et al* [27] (2007) | Dimethyl Dioxirane | (10% Oxone, 10% Acetone, 5% Sodium Bicarbonate) | 30 m submersion | 1 | 6 | Intact | 3 | Physical Changes | Oxidised metal parts |
| | | | | | | | | Odor | White residue accumulated on FFRs |
| | | | | | | | | Presence of Toxic Chemical Residues | Left distinct odor on FFRs |
| | | | | | | | | | Retained in quantity by all 6 FFRs |

(*Continued*)

**Table 4.** (Continued)

| Authors | Variables of Decontamination Methods | | | | Variables of FFRs | | | Results | |
|---------|------------|---------------|----------|----------------------------|------------------------------|----------------------|------------|------------------------|-------------------|
| | Disinfectant | Concentration | Duration | No. of Decontamination Cycles | Total no. of Models used | Part of FFR exposed | Replicates | Parameters Assessed | Summary of Results |
| **MISCELLANEOUS METHODS** | | | | | | | | | |
| Heimbuch *et al* [18] | NaOCl (Bleach) wipes | 0.9% | Surface Cleaning of outer and inner layers | 3 | 3 | Intact | 3 | Microbicidal Efficacy | 3–5 log reduction of *S. aureus* in the presence of mucin |
| | | | | | | | | Filter Efficiency | Mean particle penetration was <5% |
| | | | | | | | | Mucin removal | No mucin detected, likely due to interference in measurement assay by NaOCl |
| Heimbuch *et al* [18] | BAC wipes | | Surface Cleaning of outer and inner layers | 3 | 3 | Intact | 3 | Microbicidal Efficacy | >4 log reduction of *S. aureus* in the presence of mucin in most FFR samples |
| | | | | | | | | Filter Efficiency | Mean particle penetration was <5% but more than Bleach |
| | | | | | | | | Mucin removal | Removal efficiency ranged from 21.47–76.41% but was poorer than inert wipes |
| Heimbuch *et al* [18] | Inert wipes | | Surface Cleaning of outer and inner layers | 3 | 3 | Intact | 3 | Microbicidal Efficacy | No antibacterial activity |
| | | | | | | | | Filter Efficiency | Mean particle penetration was <5% |
| | | | | | | | | Mucin Removal | Removal efficiency ranged from 21.47%-76.41% and better than BAC wipes |

**ABBREVIATIONS: FFR:** Filtering Facepiece Respirator, **H₂O₂:** Hydrogen Peroxide, **m:** Minute, **NaOCl:** Sodium Hypochlorite, **NIOSH:** National Institute of Occupation Safety & Hygiene, **g/L:** Gram/Liter, ***S. aureus:*** *Staphylococcus aureus*, BAC: Benzalkonium Chloride

effect was not pronounced after undergoing multiple cycles of decontamination [14, 23]. Whether it is a model specific issue or not should be evaluated in future studies. In terms of microbicidal efficacy, $\geq 4 \log_{10}$ reduction of enveloped viruses was demonstrated for both methods [15, 17, 20, 26]. We are of opinion that these methods are low cost, easily doable in any setting, but require more validation in terms of other respirator models and cycles of decontamination, in future studies. MGS method is particularly suitable for implementation by individuals at home and smaller healthcare settings. Sparking due to placing metallic components in microwave has been a concern but it has not been noticed in MGS method [14].

Few studies were done on Dry heat as a modality to reprocess FFRs [16, 22, 28, 29]. Physical degradation of the respirators was noted, in varying degree, with these methods using Microwave (MGI), Hot air oven (DHO) and traditional electric rice cooker (TERC). Of these, TERC has shown to be microbiologically efficacious against *B. subtilis* spores and preserve physical architecture and filter efficiency of the respirators in limited studies conducted using it [28, 29]

We opine that the literature is insufficient to either recommend or refute dry heat as a method of reprocessing for FFRs.

Ethylene oxide (EO) and Hydrogen peroxide ($H_2O_2$) are ideally suited for reprocessing of temperature sensitive articles hence, their use for reprocessing N95-FFRs is particularly promising. They have been evaluated as a reprocessing method for N95-FFRs simultaneously in 4 studies [14, 16, 22, 27] in which, FFRs were exposed to EO and $H_2O_2$ (HPGP) in their respective sterilizers for standard cycling conditions, as described in Table 3. In addition, Viscusi *et al* [22] evaluated vaporized $H_2O_2$ (HPV) generated in a commercial, automated vapor generator (BIOQUELLⓇ). FFR models were not disclosed in any of these studies. The studies found that EO performed suitably in maintaining the physical architecture and filtration efficiency of the respirators however microbicidal efficacy, user acceptability and effect of respirator fit on N-95 FFRs were not evaluated in any study. Furthermore, a study by Salter *et al* [27] found possible carcinogen and mutagen, 2-hydroxyethyl acetate (HEA) on FFRs which had undergone EO sterilization. Hence, this method cannot be recommended for reprocessing of N95-FFRs due to safety concerns and improving the safety profile of EO by increasing aeration duration post-sterilization can be explored in future studies.

Hydrogen peroxide provides microbicidal activity by way of generating free radicals and its degradation products are safe. In 3 studies, where HPGP was evaluated, no significant physical changes on the FFRs were noted [14, 16, 22] but one study [14] noted degradation in filter efficiency of 25% (9/36) respirators. However, this effect was not noted when FFRs were treated with vaporized form [22, 41]. In a commercial evaluation done for FDA by Batelle Institute on Clarus C HPV generator (BIOQUELLⓇ) in 2016, no filter degradation was noted on 3M1870 FFR even after undergoing 50 cycles of decontamination [41]. This system has been granted emergency use authorization (EUA) by FDA, after COVID-19 pandemic, for reprocessing N95-FFRs [42]. Concerns have been raised regarding throughput of HPGP as in a study authors noticed cycles were aborted in STERRADⓇ Sterilizer whenever >6 FFRs were placed [27]. This could be due to presence of cellulose in the straps of the respirators leading to absorption of $H_2O_2$ [27]. Prior to 2020, no study, in published literature, had evaluated microbicidal efficacy of $H_2O_2$ on FFRs, but recently, Fisher *et al* [43] found it effective in removing SARS-CoV-2 from N95-FFRs. Furthermore, Batelle report, also showed 6 log reduction of *Geobacillus stearothermophilus* spores on FFRs which underwent reprocessing by HPV [41]. Overall, Hydrogen peroxide in gaseous form is a suitable option for reprocessing N95-FFRs but it needs to be evaluated rigorously for other parameters such as respirator fit and also against other N95-FFR models. However, at present its availability is restricted to limited resource rich settings.

Submersion of FFRs in liquid disinfectants is a simple method of decontaminating them. Bleach was the most frequently evaluated liquid disinfectant for reprocessing of FFRs, being evaluated in 9 studies [14, 16, 18, 22, 25–29] of which, 1 used disinfectant wipes [18]. Exposure to bleach caused physical changes in the FFRs in terms of being stiff, mottled and tarnishing of metallic nosepiece [14, 16, 18, 22]. Offensive odor from FFRs was noticed in most studies [14, 16, 27]. Furthermore, chlorine release has been noted when respirators were exposed to moisture, raising concerns regarding the safety of this method if a person breathes through it [16, 27]. Though it has been found to have no significant degradation in the filter quality of the FFRs [14, 16, 18, 22] and have excellent microbicidal efficacy [18, 25, 26, 29], FFRs decontaminated by bleach are not safe.

Liquid Hydrogen peroxide (LHP) in 3% concentration was able to preserve filter efficiency & physical architecture [14, 22] of the N95-FFRs and was devoid of any toxic residues post-exposure [27]. Alcohols (Ethanol and Isopropyl alcohol) have also been evaluated in 3 studies, but they are known to significantly degrade the filter efficiency due to removal of electrostatic

charges from the electret media [22, 28, 29]. Similarly, soap & water degraded the filter efficiency, as noted in a study [22].

We found that UVGI was the most widely evaluated reprocessing method, being evaluated for 23 different known FFR models. Nine known FFR models preserved their filter efficiency and 18 known FFR models achieved adequate microbicidal efficacy after undergoing reprocessing by UVGI method. However, the same FFR model: Parameter combination for UVGI was not evaluated in more than two studies. Six known FFR models were reprocessed by MGS [17, 21, 23] & MHI [21, 23] methods. Except for 3M1870, as discussed previously, none of the FFRs showed physical changes after undergoing reprocessing. In none of the studies which evaluated Gaseous chemical methods, identity of FFR models was disclosed. Thus, we suggest that future studies should include multiple known FFR models while evaluating a reprocessing method as compatibility of the FFR with the reprocessing method is of paramount importance.

A summary assessment of the body of literature, published prior to 2020, on reprocessing of N95-FFRs has been provided in Fig 2. However, the findings of this systematic review and opinion of the authors should be assessed in light of limited literature available on this topic, prior to 2020. Furthermore, readers should also consider the variability in exposure variables of the reprocessing methods and methodological variabilities in the evaluated parameters within and between reprocessing methods. For example, to evaluate microbicidal efficacy, studies have used different categories of micro-organisms and growth parameters accordingly while few included additional soiling challenges to mimic micro-organisms in human secretions. Some parameters were evaluated only in few studies such as odor, wear comfort, and donning ease were evaluated objectively only in 1 study [21], respirator fit in 2 studies [21, 23] and chemical safety in 1 study [27]. Hence, changes in these parameters which are not studied much, nevertheless are important, should be the focus of future studies. We didn't do a meta-analysis as the number of studies done to evaluate a particular parameter for a reprocessing method were few and heterogeneous in terms of both exposure & methodological variables.

As we write this review, a large body of literature on reprocessing of N95-FFRs has been already published [43–57], but when we did literature search, only few studies were published [44, 45, 49, 57] and majority were in preprint, non-peer reviewed versions. Hence, in this systematic review, we only included studies which were published prior to COVID pandemic. This review may help administrators, infectious disease specialists and infection control personnel to formulate policies for effective utilization of single use, N95-FFRs to prevent respiratory transmission of SARS-CoV-2 as well as other airborne pathogens. It will help researchers to find existing knowledge gaps in respirator reprocessing techniques and help them to design future studies. Furthermore, manufacturers may find it useful by knowing existing limitations and work their way around by developing new respirator material or design, more amenable to commonly available reprocessing techniques.

## Conclusions

We found that published literature on evaluation of reprocessing methods of FFRs was scant, prior to COVID pandemic. Physical methods of decontamination, such as using heat or radiation, were the most commonly evaluated methods for reprocessing of FFRs. Majority of studies evaluated either physical changes or effect on filter efficiency of respirators after undergoing decontamination and the microbicidal efficacy of the decontamination method. Only few studies evaluated the effect of decontamination methods on respirator fit or their chemical safety profile. We found that there was a lot of heterogeneity amongst the studies regarding the exposure variables of UVGI method, used respirator models and methodology to evaluate

microbicidal efficacy in terms of challenge micro-organisms, method of exposure of challenge micro-organism to FFRs, use of a soiling challenge and evaluated parameters.

We found that UVGI was the most commonly evaluated method in the published literature, prior to 2020 and it ticks all the boxes required for an ideal reprocessing method for N-95 FFRs. However, doses of UV-C irradiation which can achieve satisfactory microbicidal efficacy needs to be determined specifically for each FFR model. Majority of heat-based methods caused physical changes in the respirators, in varying degree, but adequately removed viral micro-organisms from the surface of FFRs without compromising filter efficiency, even after undergoing multiple cycles of decontamination. In particular, MGS method had extremely short cycle time & seems easy to implement in any setting. Few studies evaluated gaseous chemical methods such as EO and Hydrogen peroxide & found that filter efficiency of FFRs was maintained. However, safety concerns were raised on reusing FFRs which underwent reprocessing by EO, in the only study evaluating it.

To summarize, reusing N95-FFRs is need of the hour due to COVID-19 pandemic. Choosing a reprocessing method for FFR decontamination requires careful considerations of various factors such as physical changes, respirator fit, filter efficiency and chemical safety profile, besides being microbiologically efficacious. Furthermore, compatibility of reprocessing method with the FFR models used in a setting, duration of reprocessing cycle and costs involved make it an extremely complex decision for the infection control personnel and administrators. Presently, promising technologies which need to be evaluated rigorously include UVGI, HP, MGS & MHI. Though, emergency use approvals have been given to Hydrogen Peroxide STERRAD® Gas Plasma Sterilizer and BIOQUELL® Clarus C HPV generator, their presence is extremely limited worldwide, particularly in LMICs. Finding a suitable reprocessing method for N95-FFRs is also important from the perspective of infection control against airborne pathogens in LMICs, such as *Mycobacterium tuberculosis*. MGS and MHI have shown to be efficacious against enveloped viruses and not compromise the filter efficiency up to 3 cycles of decontamination, in multiple studies. Of them, MGS has an extremely short cycle and should be considered for emergency implementation in resource limited settings.

## Supporting information

**S1 Table. PRISMA checklist.**
(DOCX)

**S2 Table. Search strategy.**
(DOCX)

**S3 Table. Results of quality assessment & risk bias of included studies (after inter-author agreement).**
(DOCX)

**S4 Table. Summary of various reprocessing parameters evaluated for specific FFR models (where disclosed in included studies) by various reprocessing methods.**
(DOCX)

## Author Contributions

**Conceptualization:** Ayush Gupta.

**Data curation:** Diptanu Paul, Anand Kumar Maurya.

**Formal analysis:** Diptanu Paul, Ayush Gupta, Anand Kumar Maurya.

**Methodology:** Diptanu Paul, Ayush Gupta.

**Software:** Diptanu Paul.

**Validation:** Anand Kumar Maurya.

**Writing – original draft:** Diptanu Paul, Anand Kumar Maurya.

**Writing – review & editing:** Diptanu Paul, Ayush Gupta.

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
