## [Decision Letter · Decision Letter 0]

14 Sep 2020

PONE-D-20-23922

Exploring options for reprocessing of N95 Filtering Facepiece Respirators (N95-FFRs) amidst COVID-19 pandemic: a systematic review

PLOS ONE

Dear Dr. Gupta,

Thank you for submitting your manuscript to PLOS ONE. After careful consideration, we feel that it has merit but does not fully meet PLOS ONE’s publication criteria as it currently stands. Therefore, we invite you to submit a revised version of the manuscript that addresses the points raised during the review process.

We look forward to receiving your revised manuscript.

Kind regards,

Amitava Mukherjee, ME, Ph.D.

Academic Editor

PLOS ONE

Journal Requirements:

Reviewers' comments:

Reviewer's Responses to Questions

**Comments to the Author**

1. Is the manuscript technically sound, and do the data support the conclusions?

Reviewer #1: Partly

2. Has the statistical analysis been performed appropriately and rigorously? 

Reviewer #1: N/A

3. Have the authors made all data underlying the findings in their manuscript fully available?

Reviewer #1: Yes

4. Is the manuscript presented in an intelligible fashion and written in standard English?

Reviewer #1: Yes

5. Review Comments to the Author

Reviewer #1: Thank you for the opportunity to review this manuscript. The authors have performed an extensive literature search and review on N95 make re-processing. The topic is pertinent in the current setting, although rapidly changing. The authors have mostly stuck to peer-reviewed content, which is not always the case for studies at this time, but is beneficial to their work. The manuscript is very comprehensive, perhaps a little too much for a topic that they comment on as inadequate and sparse.

My concerns are:

• The comments “full set of personal protective equipment (PPE) including gowns, gloves, facemasks, face-shields or goggles and respirators for their protection during patient care” and “HCWs need to be protected against bioaerosols at all costs, which at minimum, is offered by use of N95 Filtering Facepiece Respirator (N95-FFR)” are misleading. In fact, citation (4) by the authors states that N95s are only needed during aerosol generating procedures. The authors should clarify this in the introduction.

• “removes > 95% particles of around 300 nm” – there is an official definition of what an N95 filters, the authors should use that

• The statement “No independent study prior to 2020 has evaluated microbicidal efficacy of H2O2 on FFRs” is not correct, see Fischer et. al 2020, among others

• The discussion is far too long, essentially putting the tables into text. As such:

• Paragraph 1 of the discussion is off topic and tangential opinion, it should be removed

• The paragraph that starts “A typical N95-FFR consists of facepiece…” was not material analysed by the literature search and should be removed

• The statement “N95-FFRs are difficult to decontaminate owing to the porous nature of the main body and electrostatically charged nature of electret media” needs a reference, this is debatable

• The discussion needs to be reduced to key points and synthesis, with a cohesive analysis of the literature. At least two pages could be removed.

• The conclusion doesn’t provide any conclusion regarding the review, but comments on the need for the review and then, again, summarizes the general concepts. There is no analysis here. There is also a dichotomy here, with one sentence stating a need for solution for low income countries (which I agree with), to the very next sentence stating an urgent need to study UV and HPV (not low income country solutions, by the authors own admission).

• My biggest concern here is that this really isn’t a systematic review. It attempts to be, and provides the needed supplementary info (i.e. PRISMA diagram) but doesn’t provide any significant analysis. This is really a narrative review with an extensive literature search.

Minor

• The virus name is actually “SARS-CoV-2” not “SARS-COV-2”

• Figure 2 legend “plotted against the reprocessing method” – not sure what the authors mean, it is a table, nothing is “plotted”

6. PLOS authors have the option to publish the peer review history of their article (what does this mean?). If published, this will include your full peer review and any attached files.

Reviewer #1: No

---

## [Author Response · Author response to Decision Letter 0]

13 Oct 2020

Comment 1: Section: Introduction

The comments “full set of personal protective equipment (PPE) including gowns, gloves, facemasks, face-shields or goggles and respirators for their protection during patient care” and “HCWs need to be protected against bioaerosols at all costs, which at minimum, is offered by use of N95 Filtering Facepiece Respirator (N95-FFR)” are misleading. In fact, citation (4) by the authors states that N95s are only needed during aerosol generating procedures. The authors should clarify this in the introduction.

Author’s Reply: The authors have suitably modified the sentence as suggested by the reviewers in the Introduction section (Page 4, Paragraph 1).

Comment 2: Section: Introduction

removes > 95% particles of around 300 nm” – there is an official definition of what an N95 filters, the authors should use that

Author’s Reply: The authors have modified the sentence and provided the reference of the official definition (Reference 7). In previously submitted manuscript, reference 5 & 7 were duplicated, now the new reference is given in place of Reference 7 (Page 4, Paragraph 1).

Comment 3: Section: Discussion

The statement “No independent study prior to 2020 has evaluated microbicidal efficacy of H2O2 on FFRs” is not correct, see Fischer et. al 2020, among others.

Author’s Reply: Fischer et al (2020) have referenced Batelle report which evaluated microbicidal efficacy of Hydrogen peroxide for decontamination of N95-FFRs. It was a third party “commercial” evaluation of Clarus C Hydrogen Peroxide vapor generator done by Batelle Institute on behalf of FDA. This wasn’t an independent study published in scientific domain. Hence, we are of the opinion that the statement is correct. Furthermore, we have referenced findings of Batelle report in the Discussion section of the manuscript, where deemed suitable (Page 42, Paragraph 1).

Comment 4: Section: Discussion

The discussion is far too long, essentially putting the tables into text. As such:

i. Paragraph 1 of the discussion is off topic and tangential opinion, it should be removed

Author’s Reply: In our opinion, this paragraph is important as it highlights the chronology of the scientific progress made in finding a suitable reprocessing method for enabling reuse of FFRs. We humbly opine that had such a method been found in the previous decade, the crises of FFRs wouldn’t have been of this humongous proportions during COVID-19 pandemic. In late to 2000s to late 2010s, there was scientific thrust in finding such a reprocessing method which got derailed in the mid to late 2010s contributing partly to the current crises of shortage of FFRs. 

ii. The paragraph that starts “A typical N95-FFR consists of facepiece…” was not material analysed by the literature search and should be removed.

Author’s Reply: The paragraph has been removed as suggested by the reviewers (Omitted).

iii. The statement “N95-FFRs are difficult to decontaminate owing to the porous nature of the main body and electrostatically charged nature of electret media” needs a reference, this is debatable

Author’s Reply: The statement & paragraph containing it has been removed in the revised Discussion (Omitted).

iv. The discussion needs to be reduced to key points and synthesis, with a cohesive analysis of the literature. At least two pages could be removed.

Author’s Reply: Discussion has been extensively modified as suggested by the reviewers. 

Comment 5: Section: Conclusion

The conclusion doesn’t provide any conclusion regarding the review, but comments on the need for the review and then, again, summarizes the general concepts. There is no analysis here. There is also a dichotomy here, with one sentence stating a need for solution for low income countries (which I agree with), to the very next sentence stating an urgent need to study UV and HPV (not low income country solutions, by the authors own admission).

Author’s Reply: The conclusions are now modified, hopefully to the reviewer’s satisfaction. The first 2 paragraphs summarize the findings of the study whereas the last paragraph gives general concepts and way for the future (Pages 45-46).

Comment 6: Section: 

My biggest concern here is that this really isn’t a systematic review. It attempts to be, and provides the needed supplementary info (i.e. PRISMA diagram) but doesn’t provide any significant analysis. This is really a narrative review with an extensive literature search.

Author’s Reply: We wholeheartedly accept the criticism of the reviewer. However, we will like to counter on the following points:

1. Systematic reviews address a specific research question using explicit methodology of collecting, selecting and appraising studies and synthesizing the results qualitatively or quantitatively, as appropriate. 

2. Our review appears as a narrative review because it addresses a broad research question and the outcome of the included studies is discussed more in “Qualitative terms” rather than quantitatively.

3. We are of the opinion that the included studies varied extensively in terms of heterogeneity hence discussion in quantitative terms will not do justice to the essence of the article. Hence, qualitative results of included studies were taken into account in discussion and Quantitative details are provided in the Tables.

4. Overall the studies have not shown conflicting results between themselves for a particular reprocessing method despite being extremely heterogeneous. Hence the findings appear as General concepts, when in fact they are qualitative synthesis of the published literature on the topic. Wherever, conflicting results for a particular reprocessing method have been observed amongst studies, we have made an attempt to discuss.

5. Hence, as suggested by the reviewer, we have revised the Discussion & Conclusion sections extensively focusing now more on the qualitative findings pertaining to the review, hopefully to the reviewer’s satisfaction.

Minor

Comment 7: 

The virus name is actually “SARS-CoV-2” not “SARS-COV-2”

Author’s Reply: This has been corrected in all the relevant places

Comment 8: 

Figure 2 legend “plotted against the reprocessing method” – not sure what the authors mean, it is a table, nothing is “plotted”

Author’s Reply: The legend has been suitably modified

---

## [Decision Letter · Decision Letter 1]

27 Oct 2020

PONE-D-20-23922R1

Exploring options for reprocessing of N95 Filtering Facepiece Respirators (N95-FFRs) amidst COVID-19 pandemic: A systematic review

PLOS ONE

Dear Dr. Gupta,

Thank you for submitting your manuscript to PLOS ONE. After careful consideration, we feel that it has merit but does not fully meet PLOS ONE’s publication criteria as it currently stands. Therefore, we invite you to submit a revised version of the manuscript that addresses the points raised during the review process.

Your manuscript can be accepted subject to minor revisions as detailed by the reviewer. Please check the term "systematic" in the title, and consider whether it may be changed to "Narrative" review.

We look forward to receiving your revised manuscript.

Kind regards,

Amitava Mukherjee, ME, Ph.D.

Academic Editor

PLOS ONE

Reviewers' comments:

Reviewer's Responses to Questions

**Comments to the Author**

1. If the authors have adequately addressed your comments raised in a previous round of review and you feel that this manuscript is now acceptable for publication, you may indicate that here to bypass the “Comments to the Author” section, enter your conflict of interest statement in the “Confidential to Editor” section, and submit your "Accept" recommendation.

Reviewer #1: (No Response)

2. Is the manuscript technically sound, and do the data support the conclusions?

Reviewer #1: Yes

3. Has the statistical analysis been performed appropriately and rigorously? 

Reviewer #1: Yes

4. Have the authors made all data underlying the findings in their manuscript fully available?

Reviewer #1: Yes

5. Is the manuscript presented in an intelligible fashion and written in standard English?

Reviewer #1: Yes

6. Review Comments to the Author

Reviewer #1: >> The statement “No independent study prior to 2020 has evaluated microbicidal efficacy

of H2O2 on FFRs” is not correct, see Fischer et. al 2020, among others.

Author’s Reply: Fischer et al (2020) have referenced Batelle report which evaluated

The authors must have the wrong Fischer et al. (2020) – I am referring to 1. Fischer, R. J., (2020) Effectiveness of N95 Respirator Decontamination and Reuse against SARS-CoV-2 Virus. Emerging Infect. Dis. 10.3201/eid2609.201524. This article does not reference any other article, but does the experiments themselves. This is but one example. It is actually very important to the manuscript, as this is now the most common method for commercial decontamination.

The discussion is still too long and re-states the results. I encourage the authors to think of how to discuss the results, not re-state them.

7. PLOS authors have the option to publish the peer review history of their article (what does this mean?). If published, this will include your full peer review and any attached files.

Reviewer #1: No

---

## [Author Response · Author response to Decision Letter 1]

1 Nov 2020

To

The Editor

PLOSONE

Sir

At the outset we would like to thank the Reviewer(s) for critically reviewing our manuscript. Here is our point by point rebuttal to the suggestions/ concerns raised by the Reviewer:

COMMENT 1: SECTION: TITLE

Please check the term “systematic” in the title, and consider whether it may be changed to “Narrative review.

Author’s Reply: In our humble opinion, a “Systematic review” differs from a “Narrative review” in the following points:

a. The research question is well defined.

b. There is clearly defined criteria for selection for articles published in literature

c. Methods of extraction of and synthesis of data are explicit

d. The critical appraisal of the quality of included studies is explicitly done

In summary, systematic reviews differ from narrative reviews in their “Methodology” rather than the “Results”. We are of the opinion that our review ticks all the boxes in this regard. Furthermore the protocol of the article is also registered with the International prospective register of systematic reviews (PROSPERO) beforehand. 

We have already explained that the results of our article are synthesized and discussed more in the qualitative terms rather than quantitative terms due to the heterogeneity in the methodologies of various articles who have researched this topic. Hence, we humbly request that we should persist with the word “Systematic” in the title.

COMMENT 2: SECTION: DISCUSSION

The authors must have the wrong Fischer et al. (2020)- I am referring to 1. Fischer R.J., (2020) Effectiveness of N95 Respirator Decontamination and Reuse against SARS-CoV-2 Virus. Emerging Infect Dis 10.3201/eid2609.201524. This article doesn’t reference any other article , but does the experiments themselves. This is but one example. It is actually very important to the manuscript, as this is now the most common method for commercial decontamination. 

Author’s Reply: We had cited the same study which the reviewer is referring to but it was a preprint version which we cited (Reference 43). Our statement in previous rebuttal “Fischer et al (2020) have referenced Batelle report which evaluated microbicidal efficacy of Hydrogen peroxide for decontamination of N95-FFRs”was based on this pre-print version but their comment about Batelle report are now removed from the final print when this article is published as “Letter” in Emerging Infectious Disease, which the reviewer is referring to, hence the confusion. Thus, we have now modified the reference of this article (No. 43) in our manuscript to final print version from the originally given pre-print version.

In the first cycle of peer review, the reviewer had commented “The statement- No independent study prior to 2020 has evaluated microbicidal efficacy of H2O2 on FFRs - is not correct, see Fischer et al 2020, among others.” We modified it in our article as “Prior to 2020, no study, in published literature, had evaluated microbicidal efficacy of H2O2 on FFRs, but recently, Fisher et al[43] found it effective in removing SARS-CoV-2 from N95-FFRs”. Our statement in the manuscript is correct probably the phrase “Prior to 2020” is ignored. This study by Fischer et al is published online in June 2020 (preprint version) & final version in September 2020. They evaluated the efficacy of VHP (Vaporized Hydrogen Peroxide) on N95 respirators contaminated with SARS-CoV-2 virus, a virus which was non-existent prior to 2020, so how can our statement be deemed as incorrect as we are specifically talking about research done Prior to 2020.

Furthermore, it is said that such inaccuracies are an example. We humbly request the reviewer to provide specific statements which are inaccurate and require correction.

Comment 3: Section: Discussion

The discussion is still too long and re-states the results. I encourage the authors to think of how to discuss the results, not re-state them. 

Author’s Reply: We humbly request the reviewer’s criticism in terms of 2 things: length of discussion and re-stating the results in discussion, but kindly consider this:

a. The discussion appears lengthy because there are multiple methods which have been evaluated in the published literature for reprocessing of FFRs. We believe that all these methods needs to be discussed individually in terms of their variables, pros, and cons so that their suitability in reprocessing of FFRs in a particular setting can be assessed.

b. Regarding, the re-stating of results in discussion section, we believe that some form of re-stating of the result is required to initiate discussion of a particular issue. 

Hence, taking the criticism in our stride, we have once again attempted to reduce the length of the discussion, hopefully to the reviewer’s satisfaction. Overall word count of our manuscript text (without abstract, tables/ figures and their legends, references) is 4633, which in our opinion is usual for such articles. 

Dr. Ayush Gupta

Assistant Professor

Department of Microbiology

AIIMS Bhopal

---

## [Editor Report · Decision Letter 2]

4 Nov 2020

Exploring options for reprocessing of N95 Filtering Facepiece Respirators (N95-FFRs) amidst COVID-19 pandemic: A systematic review

PONE-D-20-23922R2

Dear Dr. Gupta,

We’re pleased to inform you that your manuscript has been judged scientifically suitable for publication and will be formally accepted for publication once it meets all outstanding technical requirements.

Kind regards,

Amitava Mukherjee, ME, Ph.D.

Academic Editor

PLOS ONE
---

## [Editor Report · Acceptance letter]

9 Nov 2020

PONE-D-20-23922R2 

Exploring options for reprocessing of N95 filtering facepiece respirators (N95-FFRs) amidst COVID-19 pandemic: A systematic review 

Dear Dr. Gupta:

I'm pleased to inform you that your manuscript has been deemed suitable for publication in PLOS ONE. Congratulations! Your manuscript is now with our production department. 

Kind regards, 

on behalf of

Professor Dr. Amitava Mukherjee 

Academic Editor

PLOS ONE